# Quality of Life Changes in Acute Coronary Syndromes Patients: A Systematic Review and Meta-Analysis

**DOI:** 10.3390/ijerph17186889

**Published:** 2020-09-21

**Authors:** Billingsley Kaambwa, Hailay Abrha Gesesew, Matthew Horsfall, Derek Chew

**Affiliations:** 1College of Medicine and Public Health, Flinders University, Adelaide 5042, Australia; billingsley.kaambwa@flinders.edu.au (B.K.); matthew.horsfall@sa.gov.au (M.H.); derek.chew@flinders.edu.au (D.C.); 2Epidemiology, College of Health Sciences, Mekelle University, Mekelle 231, Ethiopia

**Keywords:** quality of life, percutaneous coronary intervention, coronary artery bypass grafting, physician therapy, acute coronary syndrome, systematic review

## Abstract

There is little up-to-date evidence about changes in quality of life following treatment for acute coronary syndrome (ACS) patients. The main aim of this review was to assess the changes in QoL in ACS patients after treatment. We undertook a systematic review and meta-analysis of quantitative studies. The search included studies that described the change of QoL of ACS patients after receiving treatment options such as percutaneous coronary intervention (PCI), coronary artery bypass grafting (CABG) and medical therapy (MT). We synthesized findings using content analysis and pooled the estimates using meta-analysis. We used the PRISMA guidelines to select and appraise the studies and report the findings. Twenty-nine (29) articles were included in the review. We found a significant improvement of QoL in ACS patients after receiving treatment. Particularly, the meta-analytic association found that the mean QoL of patients diagnosed with ACS was higher after receiving treatment compared to baseline (overall pooled mean difference = 31.88; 95% CI = 31.64–52.11, I^2^ = 98) with patients on PCI having slightly lower QoL gains (pooled mean difference = 30.22; 95% CI = 29.9–30.53, I^2^ = 0%) compared to those on CABG (pooled mean difference = 34.01; 95% CI = 33.66–34.37, I^2^ = 0%). The review confirmed that QoL of ACS patients improved after receiving treatment therapies although varied by the treatment options and patients’ preferences. This suggests the need to perform further study on the QoL, patient preferences and physicians’ decision to prescription of treatment options.

## 1. Introduction

There is little up-to-date evidence about changes in quality of life [1] following treatment of patients with acute coronary syndrome (ACS), a set of conditions associated with sudden reduced blood flow to the heart, common among patients in emergency departments and that often leads to hospitalization [2]. Multiple activities and steps are taken to ensure the accurate diagnosis and management, and outcomes of ACS. These include taking patient history, performing a physical examination, and reviewing electrocardiography, chest radiograph and cardiac biomarker test results [3,4]. Risk stratification is an important step within decision-making to select a coronary care or monitored step-down unit or with invasive or non-invasive strategies [5].

The choice of treatment for ACS is guided in part by multivariable risk assessment methods that help physicians provide a comprehensive risk assessment and prognosis for patients diagnosed with ACS [6]. The goal of treatments such as percutaneous coronary intervention (PCI), coronary artery bypass grafting (CABG) or medical therapy (MT) [7,8] include reduction of morbidity and mortality and the improvement of quality of life [9]. Health related quality of life (HRQoL) is a pluri-dimensional concept which used to assess the impact of health status on quality of life via multiple indicators comprising physical and emotional functioning, well-being and alleviation of symptoms [10]. Quality of life is usually assessed using self-reported quality of life instruments such as EuroQol [11]. Therefore, we should look at the performance indicators from the perspective of the healthcare system and the patients’ perspective by measuring their HRQoL evaluation. Anchah and colleagues posit that, in patients who are suffering from post-acute coronary syndrome (ACS), HRQoL is an independently predictor of mortality and morbidity [12]. Thus, the change in QoL of patients depends on the prescribed treatment in which there is a need to consider and provide objective information on QoL by physicians. This implies that guides and polices should integrate QoL measurement within treatment strategy decisions [13].

Several factors affect the post-procedural QoL such as procedural complications occurred after the procedure or treatment options, revascularization, age, gender, smoking, socioeconomic status (SES) and other chronic non-communicable diseases [14,15]. These factors determine the QoL gained after the treatment options for ACS. Several studies reported the change in QoL after ACS treatment; however, the findings reported contrasting evidence. There are studies that reported improved QoL gain when PCI alone is administered and in combination with CABG but smaller QoL gains when PCI is administered in combination with MT [16]. Nevertheless, the existing evidence is not synthesized (using meta-analysis) including the most updated literature. This study reviewed and enumerated the baseline to follow-up changes in QoL in ACS patients following treatment.

## 2. Materials and Methods

### 2.1. Study Protocol

A protocol for this study has been published elsewhere [17]. Although the protocol initially planned to assess two outcomes of which the primary outcome was the impact of QoL on physicians’ decision making in relation to therapeutic prescription and secondary outcome was changes in QoL among ACS patients after treatment, we found no study for the primary outcome which fulfil the eligibility criteria and methodological quality assessment score. Thus, this review only addressed the secondary outcome.

### 2.2. Population

The systematic review included studies which reported on baseline to follow-up changes in QoL following treatment for ACS.

### 2.3. Search Strategy

We conducted three steps to perform the search strategy. Initially, we undertook a limited search through Google scholar to develop key terms for our pre-defined concepts namely: predictors, factors, quality of life, or life quality for concept 1 [1]; physician’s therapy, percutaneous coronary intervention, PCI, angiography, coronary artery bypass grafting, CABG and revascularization for concept 2 (treatment options for ACS); and acute coronary syndrome, ACS, coronary heart disease, myocardial infarction, MI or heart infarction for concept 3 (study population).

We carried out a full systematic search (Appendix A) using all keywords and index terms described above across five databases namely Medical Literature Analysis and Retrieval System Online, Public or publisher MEDLINE (PubMed), Cumulative Index to Nursing and Allied Health Literature (CINAHL), Scopus and Web of Sciences. The three concepts above were connected by ‘AND’ to run the full search strategy. We then screened the titles and abstracts from each database and manually checked for additional resources from the relevant titles/abstracts selected for a full-text appraisal. The schematic presentation of the search strategy using the Preferred Reporting Items for Systematic Reviews and Meta-Analyses (PRISMA) guidelines is presented in Figure 1.

### 2.4. Study Selection and Quality Assessment

Two primary but independent reviewers, HAG and BK, assessed the selected papers for methodological quality using standardized Joanna Briggs Institute [18] appraisal instruments (Appendix B). Nine questions were used to appraise descriptive and cohort studies, 10 questions for clinical trials and 11 questions for systematic review studies (Appendix B). Based on the JBI critical appraisal checklist, methodological quality scores were calculated for each article. The scores were calculated as a ratio where the numerator was the number of ‘Y’s (yes), and the denominator was the sum of ‘U’s (unclear), ‘N’s (no) and ‘Y’s. ‘NA’ (not applicable) was excluded from the ratio calculation. Any differences in the selection process were resolved through discussion. The risk of bias was assessed using the Agency for Healthcare Research and Quality (AHRQ) criteria [19]. This tool evaluates selection, performance, detection, attrition and reporting biases [20]. For each study design included in the review, this tool has separate judgement criteria to assess the risk of bias and has four values of risk of bias including high, moderate, low or unclear risk [21]. *High risk of bias* is suggested if there is significant bias demonstrated by error in study design, data analysis and reporting which invalidates the findings. *Moderate risk of bias* is indicated if there is susceptibility of bias but not enough evidence to make the study invalid. This can be demonstrated by missing data that makes it difficult to assess the limitations of the study. *Low risk of bias* is implied if the bias is low and results are valid as demonstrated by an acceptable patient allocation to the comparator groups, low attrition rate as well as appropriate outcome measurement, data analysis and reporting. *Unclear risk of bias* is assumed if the studies are reported poorly making it difficult to make a judgement.

### 2.5. Data Extraction

We considered quantitative data and used the JBI data extraction tool (Appendix C) to extract the information from, and score the results of, each selected article. We contacted five authors of primary studies to provide us with further data as some necessary data were missing or unclear. This was done via a template prepared for further data requests and sent to each author. The reviewers independently checked the data extraction results.

### 2.6. Outcomes

The review considered the following outcome, which is related to ACS patients: decision to prescribe PCI or CABG or MT, estimation of mortality risk bleeding events and change in QoL following treatment. The operational definitions and measurements of these outcomes are presented elsewhere [22,23,24]. Briefly, PCI was defined as, “a percutaneous transluminal coronary angioplasty with or without bare metal stent or drug-eluting stent (DES) placement” [23]. CABG referred to a vascular graft made to bridge the obstructions in the coronary blood vessels [24]. MT includes prescription of aspirin, statins, β-blockers, and calcium channel blockers [23]. QoL was the primary exposure in this review. The following instruments were used to define QoL: Short Form 6, 12 and 36 dimension (SF-6D, SF-12 and SF-36, respectively) [25,26,27], The MacNew Questionnaire [28], Cardiac Quality of Life Index (QLI) [28], Seattle angina questionnaire (SAQ) [29,30], Duke activity status index (DASI) [31], Nottingham health profile (NHP) [32,33] and the European quality of life-5 dimensions 3 or 5 level measure (EQ-5D-3L or EQ-5D-5L). The secondary exposures or confounders included age, sex, treatment type, duration of the treatment and ACS diagnosis.

### 2.7. Data Analysis

The analysis included both descriptive and inferential statistics. Clinical descriptive information related to any non-quantifiable clinical improvement was also extracted. Furthermore, summary findings of the included articles were presented using a table and they consisted of author, setting, study design, population, sample size, outcome, and main findings. After assessing the clinical and statistical heterogeneity, meta-analysis was also carried out to assess the association of QoL with the above-mentioned outcomes. We also checked the clinical heterogeneity (differences in population or treatment protocol) [34] of included studies to determine their suitability for inclusion in the meta-analysis.

To assess statistical heterogeneity, standard Chi-square and I^2^ tests were used with a *p* value < 0.05 used as a cut off for determining statistical significance. Meta-analysis was considered appropriate if the I^2^ value of the included study was below 85%, and if at least two studies reported the outcome and exposure of interest [35]. To calculate effect sizes, a Mantel Haenszel statistical method with, forest plots, was used to graphically display the relationship between QoL and the outcomes. As the studies used a variety instruments with different dimensions to measure QoL, we standardized the values for all scores. We calculated a pooled mean difference and 95% confidence intervals (CI) using random or fixed effect meta-analysis as appropriate and while considering the degree of heterogeneity [35]. We used fixed effect meta-analysis if no heterogeneity was found and random if the heterogeneity was moderate (I^2^ < 85%?) [35]. If the number of studies that reported QoL was less than five (small), fixed effect model was considered irrespective of the degree of heterogeneity [36,37]. A funnel plot was used to assess the publication bias. The meta-analysis was conducted using RevMan-5 Software [38].

## 3. Results

### 3.1. Description of Studies

A total of 6756 potential records (6744 records from the literature search and 12 from the bibliographic review) were identified. We excluded 746 duplicated records and 5919 abstracts after screening.

Figure 1 presents the PRISMA flowchart of the systematic search. A total of 91 abstracts were retrieved for full text screening where 21 articles assessed the primary outcome, 74 articles assessed the secondary outcome and four articles assessed both. Of the articles that assessed the primary outcome, 20 articles were excluded due to not reporting the outcome of interest, exposure (3) or population (9) of interest. Similarly, of the articles that assessed the secondary outcome, 33 articles were excluded due to not reporting the outcome (21), exposure (9) and population (3) of interest. Another five articles were excluded due to not having full text records (4) or records that were not in English (1). Upon further critical appraisal through peer review, seven articles which assessed the secondary outcome were excluded as (i) the outcome of interest was not measured before and/or after treatment (5), (ii) outcome compared between ACS and non-ACS (1), and (iii) ACS patients who also had chronic kidney disease (1). Finally, 29 studies were included for review of which 16 studies [15,28,29,33,39,40,41,42,43,44,45,46,47,48,49,50] assessed disease specific QoL and 13 studies [26,51,52,53,54,55,56,57,58,59,60,61,62] assessed generic QoL. Of these studies, 13 studies were RCTs, 12 were cohort studies, two were cross-sectional studies, and two were reviews.

Of the 29 studies included for review, one study [28] assessed both primary and secondary outcomes while the rest of the studies [15,26,28,29,33,39,40,41,42,43,44,45,46,47,48,49,50,51,52,53,54,55,56,57,58,59,60,61,62] assessed the secondary outcome. Therefore, we only report and discuss the secondary outcome as no adequate study was found for the primary outcome. Table 1 presents the characteristics and secondary outcomes of the included studies. Studies from Europe were overrepresented (18) follows by US/Canada. The sample size in each included study ranged from 27 to 1957, and 48,862 participants were included in total.

### 3.2. Methodological Quality

Table 2 shows the total score of each study depicting their methodological quality based on JBI appraisal instruments. As shown in the table, nine studies met all (100%) of the quality appraisal criteria, and 12 studies only missed out on one criterion. Table 3 summarizes the risk of bias of the included studies using the AHRQ criteria. The extent of risk bias in most studies was ‘*low risk*’. There were few studies with an ‘*unclear risk*’ status due to reporting inadequacies and the lack of determination of the nature of the study design. One study [28] was deemed to possibly have ‘*high risk*’ status in two criteria namely allocation concealment (selection bias) and blinding of participants (performance bias). There were disagreements in two articles (2/29) in the final screening process which provided a level of percent agreement to 93.1%.

### 3.3. Measurement of QoL

QoL was measured using different tools made up of different items, dimensions and scoring algorithms. All studies reported on the reliability and validity of QoL measurement tools. Table 1 also lists the type of instruments used to measure QoL. The SF-36 was used by the most studies (reported in 11 studies) while the EuroQoL 5 dimensions (EQ-5D), SAQ and MacNew were also used in five, four and three studies, respectively. Other tools used by the studies were the QLI, NHP, visual analogue scale (VAS), The World Health Organization quality of life tool-36 (WHOQoL36) and medical outcomes study (MOS) measure.

### 3.4. Changes in QoL among ACS Patients after Treatment

Twenty-nine (29) studies [15,26,28,29,33,39,40,41,42,43,44,45,46,47,48,49,50,51,52,53,54,55,56,57,58,59,60,61,62] assessed the status of QoL of ACS patients after receiving treatment strategies. Although QoL was measured using different instruments, most studies reported improvements in QoL of ACS patients after receiving treatment. However, the improvements vary according to dimensions of QoL measured, treatment type and duration of treatments as well as in terms of clinical and non-clinical characteristics of patients. Thus, a significant finding in this review was differed according to patients’ clinical and non-clinical characteristics: sex, age and ACS diagnosis. While almost all studies found marked improvements in QoL after treatment, there were mixed results when the results were analysed according to the domains or dimensions of the QoL instruments. For example, Blankenship et al. [63] found improvements of QoL in the majority of the dimensions such as angina frequency, bodily pain, and disease perception but not in angina stability or treatment satisfaction. Favarato et al. [26] also revealed higher improvements in physical functioning, general health, vitality and pain but not in emotional and mental health.

QoL by treatment type also showed mixed findings. The vast majority of included studies reported improvements in QoL of patients from baseline after PCI and CABG procedures. Specifically, QoL after PCI compared with CABG was better in the short-term, but similar or worse in the intermediate and longer term [26,28,29,43,50,54,58]. For example, Borkon et al. [29], showed that the QoL had improved at one-month post PCI procedure compared to CABG. However, the result was reversed at 6 and 12 months with CABG showing greater improvement in QoL than PCI. Cohen et al. [43] reported higher scores of the SAQ, SF-36 and EQ-5D for patients receiving PCI compared to those receiving CABG after one month, but the reverse after 6 months. To the contrary, Bahramnezhad et al. [52] reported reduction in QoL score three months after Percutaneous transluminal coronary angioplasty (PTCA) (*p* = 0. 04) but improvements after six months (*p* < 0.001). Wahrborg et al. [33] found similar scores of QoL in patients with and without therapeutic strategies. Tegn et al. [48] also found no statistically significant difference in QoL score between invasive strategy (IS) and conservative strategy (CS) treatments in all domains except for bodily pain.

QoL was found to differ according to patients’ clinical and non-clinical characteristics: sex, age and ACS diagnosis. Most studies included in this review found high QoL scores in males than females. For example, Azmi et al. [51] revealed that males had higher QoL than females (*p* = 0.003) with other studies such as Blankenship et al. [63], Favarato et al. [26] and Veenstra et al. [50] confirming this finding. On the contrary, Wahrborg et al. [33] found no difference by gender in both PTCA and CABG treatment strategies. Blankenship et al. [63], Veenstra et al. [50] and Yan et al. [61] characterized QoL by age. Blankenship et al. [54] revealed that older patients had higher QoL score compared to their younger comparators. Conversely, Veenstra et al. [50] found higher physical dimension scores among younger adults after invasive coronary procedures. Similarly, Yan et al. [61] also found a significant gain in two dimensions of QoL (pain and depression) for younger patients at 36 months after treatment. Azmi et al. [51] found higher QoL score in patients on STEMI compared to those on NSTEMI/UA (*p* = 0.045).

In summary, QoL improved in the vast majority of dimensions irrespective of the heterogeneity in measurement instruments. The evidence of improvement in QoL by treatment type, duration, and clinical and non-clinical characteristics was more mixed.

### 3.5. Meta-Analysis of Status of QoL among ACS Patients at Baseline and after Receiving Treatment

This meta-analysis determined if QoL of ACS patients at baseline was different to that after receiving treatments by estimating the pooled effect size. The meta-analysis is stratified for both treatments (PCI and CABG). However, only few studies were included in both subgroups due to: (i) instruments used for measuring QoL and the scoring algorithms were disparate, (ii) data of baseline or after treatment or both, and SD in several papers included in the systematic review were not reported even though efforts were made to contact authors, and (iii) some of the included papers (weighted 0% in the forest plot in Figure 2) introduced high heterogeneity (I^2^ > 85%). One source of heterogeneity for the meta-analysis was the variation in time points at which QoL was assessed from study to study. For example, Koltowski et al. [28] assessed the QoL gains at two hours and four days after treatment while Abdallah et al. [41] assessed it at two years after treatment. In this study, we compared the QoL gains between baseline (before treatment) and final assessment of the QoL after treatment (for example, at four days for Koltowski et al. [28] and at two years for Abdallah et al. [41]).

We used a fixed effects meta-analysis model because few studies (<5) were included in the meta-analysis in each group [35,36,37]. As shown in Figure 2, we excluded five studies each (weighted 0%) from the meta-analysis calculation of QoL after PCI and CBAG respectively as they introduced high heterogeneity. The meta-analytic association found that the mean QoL of ACS patients was higher after receiving any treatment compared to that at baseline (Figure 2, overall pooled mean difference = 31.88; 95% CI = 31.64–52.11, I^2^ = 98). Patients who underwent PCI had slightly lower QoL gains (pooled mean difference = 30.22; 95% CI = 29.9–30.53, I^2^ = 0%) than patients who underwent CABG (pooled mean difference = 34.01; 95% CI = 33.66–34.37, I^2^ = 0%). As the heterogeneity level of the overall pooled mean difference is significant (I^2^ test = 98%, *p* < 0.0001), this conclusion should be interpreted cautiously. Publication bias was not assessed using funnel plot as the studies included for each of treatment were very small.

QoL and its impact on the decision to prescribing ACS treatment strategies (PCI, CABG or MT) is a neglected area of research in clinical medicine. Furthermore, post-procedural QoL gain is also not updated. As a result, our study reviewed evidence on whether physicians assess patients’ QoL before choosing different therapeutic options, whether QoL impacts their prediction assessment of risk and the change in QoL following treatments. However, our review did not find any evidence of the use of QoL assessment within the treatment-strategy-decision-making process for ACS patients. This is despite studies in other disease areas emphasising the importance of, and recommending the, measuring QoL for decision making for therapeutic preferences [63,64,65,66,67,68,69,70]. For example, Sutherland and Till [63] emphasised the importance of assessing QoL at different levels of decision making: (i) at *micro-level* to help ACS patients participate in the decision making process relating to their preferred treatment/s), (ii) at *meso-level* to benefit groups at agency, institution or regional network level (e.g., teams of cardiologist may take part in developing protocols for management of ACS or small agencies working on ACS management), and (iii) at *macro-level* to benefit the population in general (e.g., the baseline data about QoL may help to the government to design population-level policy). In their review, Goodwin and colleagues [70] found that the assessment of QoL influenced clinical decision making of breast cancer patients in five of eight studies of primary breast cancer management.

Studies also demonstrated in favour of feasibility of assessment of QoL before prescribing treatment/s in patients with other chronic conditions. For example, Detmar and colleague [64] investigated the feasibility of assessing individual patients’ QoL in oncology outpatient clinic in daily practices. The study found that it took an average of 5.5 (range: 2.5–13) minutes to measure QoL using the European Organization for Research and Treatment of Cancer quality of life questionnaire (EORTC QLQ-C30) [71] implying that baseline QoL assessment could always be undertaken during the waiting times in treatment centres. In their systematic review, Goodwin and colleagues [70], however, suggested that the QoL assessment tool should be easy to fill by patients and simple to score for physicians. The study by Detmar and colleague [64] found that patient-doctor communication improved when QoL was embedded into the daily routine practice, as also supported by the systematic review from Goodwin and colleagues [70]. In an outpatient clinic setting, the assessment of QoL was additionally found to lead to better monitoring of patient care [70], improvement in patient care services [72], and improvement in physicians’ awareness of patients’ daily life and psychosocial functioning [64].

Therefore, although the need to assess QoL for decision making to prescribe preferred drugs has been previously recommended [63], there is still lack of evidence that this takes place for ACS patients in practice. There was however some evidence that QoL was predictive of adverse clinical outcomes such as in-hospital mortality and could therefore be used for risk assessment of such outcomes. In view of our findings, we highly recommend further research on the impact of QoL on decision making relating therapeutic preferences among ACS patients at *micro*-, *meso*- and *macro*-levels. Furthermore, future research should also consider determining the impact of QoL assessment on risk estimation of more varied ACS patients’ outcomes as evidence on this was limited in our review.

The other objective of this review was synthesizing the most up to date evidence on changes in QoL after treatment for ACS patients. Most of the 29 studies included in the review revealed improvement in QoL after treatment irrespective of the type of therapy. PCI provided short-term QoL improvement but CABG provided long-term QoL gain. In the review by Takousi and colleagues [49], PCI and CABG provided better QoL than MT but no statistically significant difference was confirmed between the two treatments. The main reason for the difference might be due to the differences in the tool used to assess the QoL. It is evident in the literature that using generic QoL instruments could provide a different effect size of improvements in QoL than population or disease-specific measurements as they are less sensitive than the latter [73]. Therefore, the absence of a gold standard definition for QoL could lead to significant differences in the scores of various instruments. Although evidence on the baseline-to-follow-up change in QoL is mixed, the fact that most simple treatments result in short term QoL gains and complex treatments result longer-term QoL gains should be communicated to patients. Shabason and colleagues emphasised the importance of patient engagement in the decision-making process in oncology [74].

The improvements of QoL by the type and duration of treatment, and clinical/non-clinical characteristics of patients were mixed. Takousi and colleagues [49] found that the effect size of improvements in QoL was slightly higher at one year than at three or four years after coronary revascularization (PCI/CABG) but neither age nor sex was found a predictor for QoL in this review.

There are some limitations in the present study and readers should take caution while interpreting the review results. Firstly, we are unable to make any conclusions about the impact of QoL on decision making to therapeutic prescription among ACS patients due to lack of relevant studies. Secondly, the heterogeneous nature of the included studies (mostly due to differences in instruments used for measuring QoL, disparate follow up periods and the absence of data) affected the meta-analytic association for the secondary outcome. It is possible that the small changes in HRQoL scores seen in some of the studies included in our review could have been due to treatment driven by perceived threat to life rather than direct impact on quality of life. The inclusion, within the metanalysis, of many RCTs may have however reduced this confounding. Thirdly, the lack of data of most included studies on predictors of QoL after treatment/s also prohibited us from calculating the meta-analytic association of the predictors. Furthermore, efforts to contact corresponding authors to provide unreported additional data were unsuccessful. Finally, potential articles included in the review were limited to English language and this could lead to reporting bias [75].

## 4. Conclusions

This review revealed that QoL of ACS patients improved after receiving treatments when compared with the baseline status where treatments were not provided. Furthermore, patients with PCI reported improved QoL gain in the first few months whereas patients with CABG procedures reported improved QoL gain in the later months after the procedure. This systematic review did not find any evidence to test the hypothesis regarding whether ACS patients’ QoL has an influence in physicians’ treatment decisions. There is thus need for future research to assess the role of QoL on physicians’ decisions to treatment prescriptions, and also potential risk estimations of various clinical outcomes including bleeding risks.

## Figures and Tables

**Figure 1 ijerph-17-06889-f001:**
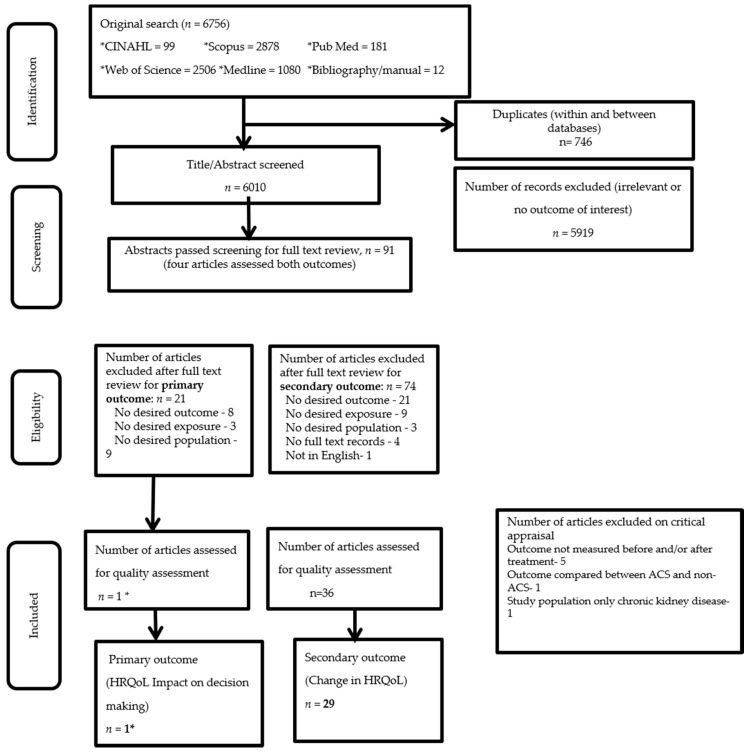
Preferred reporting items for systematic reviews and meta-analyses (PRISMA) flowchart for identification and selection of studies for inclusion in the systematic search, 2019. ** One article assessed both outcomes.*

**Figure 2 ijerph-17-06889-f002:**
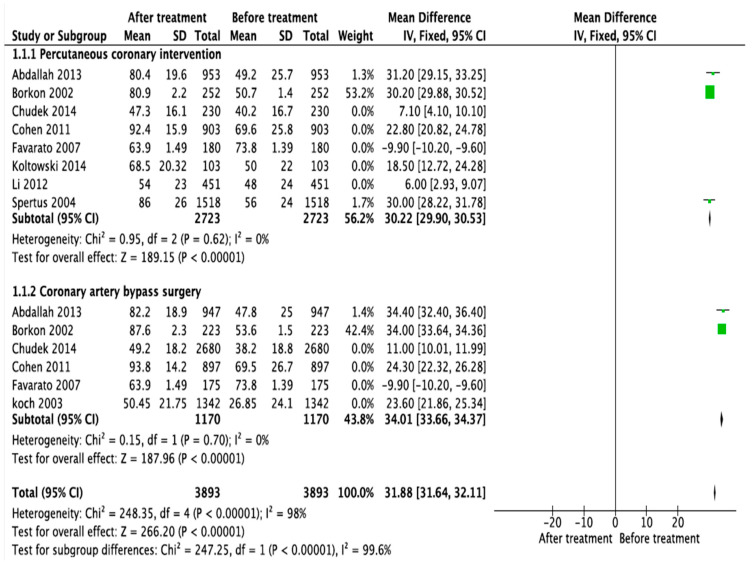
Forest plot of meta-analytic association between QoL and ACS treatments, 2019.

**Table 1 ijerph-17-06889-t001:** Characteristics of included articles (*n* = 29) ^1^.

Author	Year, Country	Sample Size	Study Design	Measurement	Summary
***Secondary outcome: QoL change after treatment options***					
Aasa et al. [39]	2010, Sweden	205	RCT	EQ-5D	QoL was assessed at baseline, one month and 12 months after primary PCI. The QoL improved one year after PCI.
Abdallah et al. [40]	2017, USA and Europe	1800	RCT	SAQ and SF-36	Health status was assessed at baseline, 1, 6, 12, 34 and 60 months after CABG or DES-PCI. Health status was improved after receiving both treatments, and further improvements were found as the follow up period increases although differences in time were observed by differences. At one month after treatment, patients assigned to DES-PCI had faster health status improvements when compared with CABG. However, the reverse was found at year after follow up— the health status of patients assigned to DES-PCI was not sustained but patients assigned to CABG found their health status improved. No differences at three years, but slight differences favouring CABG were observed.
Abdallah et al. [41]	2013, Multi- countries (18)	1900	RCT	SAQ	Health status was assessed at baseline, 1, 6, and 12 months after CABG or PCI The health status scores were improved at two years follow up for both treatments compared to baseline, but no differences in health status score beyond two years. For example, the baseline mean score of QoL for CABG and PCI groups was 47.8 and 49.2 respectively, and mean score of QoL after two years was 82.2 and 80.4 respectively.
Benzer et al. [42]	2003, Austria	267	Cohort study	MacNew	The HRQoL global and scale scores were improved after treatments. Global HRQoL mean score was -0.290 at baseline but 0.25 for medical therapy, 0.58 for PCI and 0.9 for CABG after 12 months, and P-value was significant for all changes i.e., 0.015, 0.001 and 0.001 respectively for these therapies. Post-hoc analyses showed that HRQoL improved significantly with patients assigned to CABG and PCI than medical therapy.
Borkon et al. [29]	2002, USA	495	RCT	SAQ	QoL had improved after PCI and CABG procedures Both PCI and CABG facilitated a time-dependent improvement in risk-adjusted HRQoL. HRQoL had improved after PCI than after CABG at one-month post procedure, but long term QoL such as QoL at 6 and 12 months after CABG procedure was better to a greater extent than after PCI. For example, the baseline mean score for PCI and CABG was 59.7 and 53.6, but was 70.4 and 65 at 1 month and 80.9 and 87.6 at 12 months respectively.
Cohen et al. [43]	2011, Europe/North America	1800	RCT	SAQ ^2^, and SF-36& EQ-5D ^3^	The SAQ score increased to a greater extent with CABG than with PCI at both 6 (*p* = 0.04) and 12 months (*p* = 0.03). Specifically, At 1 month, the QoL benefits of PCI over CABG was high in seven of the eight SF-36 domains and three domains of SAQ. Higher EQ-5D scores were recorded in patients who received PCI compared to those who got CABG at one month. However, very few of these differences persisted at 6 months. Conversely, the scores on the general health subscale were higher in the CABG group than in the PCI group by 12 months.
Kim et al. [44]	2013, Korea	3577	RCT	SAQ, and EQ-5D	HRQoL was measured at baseline and 30 days. The overall HRQoL improved after PCI but the angina related HRQoL improvement among patients with Non -STEMI was higher (44.2%) than those with STEMI (36.8%). For the general HRQOL, the improvement between Non -STEMI and STEMI was comparable with a mean score of 56.1 vs. 56.6 respectively.
Kim et al. [45]	2005, United Kingdom	1810	RCT	SAQ&SF-36 ^4^, and EQ-5D	Although there is an overall improvement in quality of life after treatment, early interventional strategy (IS) provided more benefit in QoL than conservative strategy (CS), mainly due to improvements in angina grade. The QoL mean score was found to be higher in patients who were treated with IS over CS at four months and one year (*p* < 0.05 for both time points), and this could possibly be due to improvements in angina symptoms. The EQ-5D, at both follow-up times, shows that 18% of patients in the CS group had a worsening of QOL related to performing usual daily activities. More patients in the CS group (20% at 4 months and 16% at 1 year) also had poorer QoL due to anxiety than in the IS group (15% at both points of time). When assessed using SF-36, the QoL score was better at one year for physical, social, and emotional role functions, and vitality and general health; and also, at four months with the exception of bodily pain and mental health. When assessed using SAQ, the mean scores were significantly higher in the IS than CS group at both points of time.
Koltowski et al. [28]	2014, Poland	103	RCT	EQ-5D, MacNew, QLI	QoL was assessed before PCI, and two hours and four days after PCI. Two hours after PCI, the mean utility score was 0.46 (±0.291): 0.60 (±0.299 for TR group and 0.32 (±0.283) for TF groups, *p* < 0.001). An improvement in QoL compared with pre-interventional evaluation was observed at two hours and four days after PCI.
Li et al. [46]	2012, China	624	Cohort study	SF-36	The QoL score improved six months after PCI treatment in all items. The overall score of physical and mental component summary were increased from 32 and 51 at baseline to 42 and 53 at six months after PCI treatment respectively.
Rinfret et al. [47]	2001, France	509	RCT	SAQ and SF-36	HRQoL improved after treatment over follow ups. HRQOL scores improved over the follow-up period for three of five domains of the SAQ (anginal frequency, disease perception and physical limitations due to heart disease; all *p* < 0.001) and for seven of eight domains of the SF-36 (all except general health perception; all *p* < 0.001).
Schenkeveld et al. [15]	2010, Holland	872	Cohort study	SF-36	The distribution of the patients with poor or good health status on 1 and 12 months post-PCI was as follows: 12% had good health status at 1 month but poor health status at 12 months, 9% had poor health status at 1 month but good health status at 12 months, 59% had good health status at both times, and 20% had poor health status at both times.
Tegn et al. [48]	2018, Norway	457	RCT	SF-36	No significant changes in QoL scores were observed between IS and CS in any of the domains of QoL. Only a small but statistically significant difference (*p* = 0.0267) was observed in bodily pain, in which the mean baseline score for IS and CS was 0.16 and 0.18 respectively and 0.16 and 0.15 a year after treatment respectively.
Takousi et al. [49]	2016	15992	Systematic review and meta-analysis	MAcNEW, SF-36, NHP, SAQ, EQ-5D, RAND-36 ^5^, WHOQOL-Brief,	Both PCI and CABG had significantly greater effects on QoL than did medication; however, the coronary revascularization (CR) procedures did not differ significantly from each other.
Veenstra et al. [50]	2004, Norway	254	Cross-sectional study	SF-36	Improvements were recorded in most dimensions such as physical and emotional role limitations and social functioning after undergoing invasive coronary procedures. Significant improvement was reported in the physical role limitation during the 2 years following invasive coronary procedures. The findings showed that young, male, and more educated patients had a higher increase in the QoL score following invasive coronary procedures.
Wahrborg et al. [33]	1999, Sweden	154	RCT	Nottingham Health Profile and a set of 12 other questions	A significant improvement in QoL in all dimensions was found when compared between the baseline and after treatments. However, no significant difference was reported when analysed by the type of treatment strategies- PTCA ^6^ and CABG. Sex was not found predictor of HRQoL even stratified aby type of treatments such as PTCA or CABG.
***Secondary outcome: Health-related or generic QoL***					
Azmi et al. [51]	2015, Malaysia	104	Retrospective cohort	EuroQol	The HRQoL generally improved from baseline to 12 months after the ACS event and this was confirmed by the change in utility scores based on Malaysian and United Kingdom (UK) tariffs. Utility scores increased from 0.75 to 0.82 (*p* = 0.014) and 0.63 to 0.72 (*p* = 0.086) when calculated using the Malaysian and UK utility tariffs, respectively. QoL was only found different by sex and diagnosis type of acute coronary syndrome. The utility score in males was higher (0.76) when compared to females’ score (0.65) (0.003) at baseline. Additionally, the utility score of patients on ST segment elevation myocardial infarction (STEMI) was higher (0.78) when compared to patients on non-STEMI (0.71) (*p* = 0.045).
Bahramnezhad et al. [52]	2015, Iran	115	Longitudinal Study	No tool described	Compared to the baseline measurement, the QoL score was reduced three months after PTCA (*p* = 0.04) but improved after six months (*p* < 0.001). No quantifiable estimates of QoL was presented
Bakhai et al. [53]	2015, Europe (14 countries)	4546	Cohort study	EQ-5D	HRQoL was measured at baseline and 12 months post PCI stratified by sex. It shows that HRQoL was improved 12 months after receiving PCI The baseline to 12 months post PCI mean (standard error) change in HRQoL VAS score was 6.92(24.37) for women and 7.86 (23.21) for men; and mean (standard error) HRQoL scores at 12 months were 70.4 (18.97) for women and 76.29 (16.94) for men).
Blankenship et al. [63]	2013, USA	---	Literature review	Ferrans and Powers Quality of Life Index, McMaster Health Index Questionnaire, SF-36, SF-12, NHP, Psychological wellbeing index, Quality of wellbeing scale, Sickness Impact Profile, Swedish HRQoL survey, DASI	In the majority of quality dimensions, the review demonstrated the improvements of QoL after PCI. No quantifiable estimates of QoL was provided as the review did not perform a meta-analysis. QoL of patients was also improved after CABG procedure compared to before. The review found that QoL was better after PCI than CABG in the first months after the procedures. However, the QoL got worse 1–5 years after PCI although no difference after longer periods. Age and sex were found the predictors of QoL. Men had higher QoL after PCI than women, and elder group of patients had higher QoL score than their young comparator. The importance of QoL issues should be considered in all aspects of PCI/CABG care from the physician’s initial assessment. In the short-term, patients may choose less complex treatment strategies (e.g., PCI) over complex ones (e.g., CABG) despite the possibility of the latter leading to a better outcome in the long term. Therefore, physicians should consider and discuss these trade-offs with their patients while prescribing treatment options. Studies have also demonstrated greater gains in QoL with an invasive strategy leading to PCI when appropriate compared with a strategy of medical therapy in acute coronary syndrome patients
Bourassa et al. [55]	2000, US/Canada	1095	Cohort study	DASI ^7^	Functional status improved 1-year after CABG (DASI score 13.5 vs. 6.0, *p* = 0.002) but the difference progressively narrowed after 5 years.
Chudek et al. [56]	2014, Poland	3220	Survey	SF-12	The improvements in quality of life was recorded regardless of treatment type. The QoL score at visit 2(after 2 months of treatment) and 3(4 months after treatment) is higher than at visits 1(baseline) for four different acute coronary syndrome management types including non-invasive, fibrinolysis, angioplasty, and stenting. For the non-invasive group, the highest change in QoL score (18.1) was in the physical health domain whereas the lowest change (10.9) was in the psychic health. For the fibrinolysis group, the highest change in QoL score (20.2) was in the physical limitation domain whereas the lowest change (14.1) was in the general health. For the angioplasty group, the highest change in QoL score (12) was in the emotional condition domain whereas the lowest change (7.1) was in the general health. For the stenting group, the highest change in QoL score (14.2) was in the physical health domain whereas the lowest change (9.8) was in the physical pain.
Favarato et al. [26]	2007, Brazil	542	RCT	SF-36	Patients showed significant improvements after receiving PCI, CABG and medical therapy after 12 years. In particular, the QoL was significantly high in physical role functioning, general health, vitality and pain domains. However, QoL was better in both CABG and PCI groups compared to medical therapy after 1 year of follow-up. However, the CABG group showed highly significant superiority over the PCI and medical therapy group in terms of vitality (*p* = 0.0024) and physical functioning (*p* = 0.0029) at 6 months and also over the medical therapy group in terms of general health at both 6 and 12 months (*p* < 0.001). Compared to women, the QoL score among men was higher in the earlier period of treatment, although the gain in the later periods, after 6 and 12 months after treatment, was progressive.
Koch et al. [57]	2003, USA	1825	Survey	DASI	The median score of DASI is higher for ACS patients two years after CABG compared to baseline score. The median baseline DASI (women, 21.5; men, 32.2; *p* = 0.001) and first follow-up scores (women, 42.7; men, 58.2; *p* = 0.001) were lower in women than in men.
Krzych et al. [62]	2009, Poland	50 ^8^	Cohort	MacNew	Although the QoL in men aged below 65 years deteriorated significantly a few days after CABG treatment, the score improved few weeks after the treatment. For example, emotional domain deteriorated shortly after CABG from 4.97 to 4.66, physical domain from 4.49 to 4.2 and social domain 4.68 to 4.47. However, the score of emotion domain improved from 5.29 to 5.96, physical domain from 4.66 to 5.42 and social domain from 4.69 to 5.65.
Sipotz et al. [58]	2013, Austria	163	Record review or registry	MacNew Health Related Quality of Life	The improvements in HRQOL score were found in the short term after PCI i.e., up to six months (*p* < 0.001) but remained stable until 2 years. Specifically, the Physical and Social dimensions showed marked improvements.
Sjoland et al. [59]	1999, Sweden	2121	Cohort study	Physical Activity Score, the Nottingham Health Profile (NHP) & Psychological General Well-being Index	The QoL life was measured at baseline, 3 months, 1 year and two year after surgery ^9^. The physical activity score at three months after treatment in both sexes was significant (3.6 for females and 3.02 for males) compared to prior surgery (4.55 for females and 4.22 for males) but the improvement thereafter was minimal. The NHP score at three months after surgery in both sexes was significant (14.3 for females and 10.8 for males) compared to prior surgery (28 for females and 19 for males) but the improvement thereafter was stable Similarly, the mean score for Psychological General Well-being Index at three months after surgery in both sexes was significant compared to prior surgery but the improvement thereafter was similar.
Spertus et al. [60]	2004, USA	1518	Cohort study	SAQ	QoL score was improved after PCI. The mean score after PCI for Physical Limitation, Angina Frequency, and Quality- of-Life domains increased by 18, 24, and 30 points, respectively (*p* < 0.0001).
Yan et al. [61]	2018, China	1957	Prospective cohort study	EQ-5D, VAS	A significant gain in benefit of HRQOL was registered in the first six months (compared to baseline) after PCI in all age groups but relatively stable thereafter. For participants age below 65 years old, the VAS score improved from 50.1 at baseline to 71.2 at 6 and 12 months each respectively and 72.9 at 36 months after treatment. For participants between 65–74 years, the VAS score improved from 51.6 at baseline to 70.9, 71.1 and 72.8 at 6, 12 and 36 months after treatment respectively. For participants older than 75 years, the VAS score improved from 52.6 at baseline to 70.5, 71.2 and 72 at 6, 12 and 36 months after treatment respectively.

^1^ RCT: randomized clinical trial; HRQoL: health related quality of life; QoL: health-related or generic quality of life; EQ-5D: The European quality of life (EuroQol)-5 dimensions; QLI: Cardiac Quality of Life Index; SAQ: Seattle angina questionnaire; SF-36: Medical Outcomes Study Short Form Questionnaire- 36; PCI: percutaneous coronary intervention; CABG: coronary artery bypass grafting; DES: drug-eluting stent. ^2^ SAQ measures disease-specific health status. ^3^ SF-36 and EQ-5D measure general health status. ^4^ SAQ and SF-36 used to measure QoL at four months and one year, EQ-5D used to measure at baseline. ^5^ RAND-36: Rand-36 item health survey. ^6^ PTCA: percutaneous transluminal coronary angioplasty. ^7^ DASI: Duke activity status index. ^8^ Men age < 65 years old were the study participants. ^9^ While interpreting the score, high score means poor quality of life whereas a low score means good quality of life.

**Table 2 ijerph-17-06889-t002:** Assessment of methodological quality of included studies (*n* = 29).

*Secondary Outcome*												
***Authors***	***Cross-Sectional Studies***											
	**Was study based on a random or pseudo-random sample?**		**Were the criteria for inclusion in the sample clearly defined?**	**Were confounding factors identified and strategies to deal with them stated?**	**Were outcomes assessed using objective criteria?**	**If comparisons are being made, was there sufficient comparison of groups?**	**Was the follow-up carried out over a sufficient period?**	**Were the outcomes of people who withdrew described and included in the analysis?**		**Were outcomes measured in reliable way?**	**Was appropriate statistical analysis used?**	**%**
Kim et al. [44]	NA		Y	N	Y	Y	NA	NA		Y	Y	83
Koch et al. [57]	NA		Y	Y	Y	Y	NA	NA		Y	Y	100
Veenstra et al. [50]	N		Y	Y	Y	Y	NA	Y		Y	Y	88
	***Cohort Studies***											
	**Is the sample representative of patients in the population?**		**Are the patients at a similar point in the course of their condition/illness?**	**Has bias been minimized in relation to selection of cases and controls?**	**Are confounding factors identified and strategies to deal with them stated?**	**Are outcomes assessed using objective criteria?**	**Was the follow-up carried out over a sufficient period?**	**Were the outcomes of people who withdrew described and included in the analysis?**		**Were outcomes measured in reliable way?**	**Was appropriate statistical analysis used?**	**%**
Azmi et al. [51]	Y		Y	NA	N	Y	Y	NA		Y	Y	86
Bahramnezhad et al. [52]	Y		Y	NA	N	Y	Y	N		Y	Y	75
Bakhai et al. [53]	Y		Y	NA	Y	Y	Y	N		Y	Y	88
Benzer et al. [42]	Y		Y	NA	N	Y	Y	N		Y	Y	75
Chudek et al. [56]	Y		Y	NA	N	Y	Y	NA		Y	Y	86
Krzych et al. [62]	N		Y	Y	Y	Y	Y	NA		Y	Y	86
Li et al. [46]	Y		Y	NA	Y	Y	Y	NA		Y	Y	100
Schenkeveld et al. [15]	Y		Y	NA	Y	Y	Y	NA		Y	Y	100
Sipotz et al. [58]	Y		Y	Y	Y	Y	Y	NA		Y	Y	100
Sjoland et al. [59]	Y		Y	Y	Y	Y	Y	NA		Y	Y	100
Spertus et al. [60]	Y		Y	Y	Y	Y	Y	NA		Y	Y	100
Yan et al. [61]	Y		Y	Y	Y	Y	Y	N		Y	Y	89
	***Clinical Trial Studies***											
	**Was the assignment to treatment groups truly random?**		**Were participants blinded to treatment allocation?**	**Was allocation to treatment groups concealed from the allocator?**	**Were the outcomes of people who withdrew described and included in the analysis?**	**Were those assessing outcomes blind to the treatment allocation?**	**Were the control and treatment groups comparable at entry?**	**Were groups treated identically other than for the named intervention?**	**Were outcomes measured in the same way for all age groups**	**Were outcomes measured in a reliable way?**	**Was appropriate statistical analysis used?**	**%**
Aasa et al. [39]	Y		NA	Y	N	Y	Y	Y	Y	Y	Y	89
Abdallah et al. [40]	Y		N	NA	N	NA	Y	Y	Y	Y	Y	75
Abdallah et al. [41]	Y		N	NA	N	NA	Y	Y	Y	Y	Y	75
Borkon et al. [29]	Y		Y	Y	N	Y	Y	Y	Y	Y	Y	90
Bourassa et al. [55]	Y		N	Y	NA	Y	Y	Y	Y	Y	Y	89
Cohen et al. [43]	Y		Y	Y	N	U	Y	Y	Y	Y	Y	80
Favarato et al. [26]	Y		U	U	Y	Y	Y	Y	Y	Y	Y	80
Kim et al. [45]	Y		Y	Y	Y	Y	Y	Y	Y	Y	Y	100
Koltowski et al. [28] *	Y		N	N	Y	N	Y	Y	Y	Y	Y	70
Rinfret et al. [47]	Y		N	NA	NA	NA	Y	Y	Y	Y	Y	86
Tegn et al. [48]	Y		Y	Y	Y	Y	Y	Y	Y	Y	Y	100
Wahrborg et al. [33]	Y		Y	Y	Y	Y	Y	Y	Y	Y	Y	100
	***Review Studies***											
	**Is the review question clearly and explicitly stated?**	**Were the inclusion criteria appropriate for the review question?**	**Was the search strategy appropriate?**	**Were the sources and resources used to search for studies adequate?**	**Were the criteria for appraising studies appropriate?**	**Was critical appraisal conducted by two or more reviewers independently?**	**Were there methods to minimize errors in data extraction?**	**Were the methods used to combine studies appropriate?**	**Was the likelihood of publication bias assessed?**	**Were recommendations for policy and/or practice supported by the reported data?**	**Were the specific directives for new research appropriate?**	**%**
Takousi et al. [49]	Y	Y	Y	Y	Y	N	Y	Y	N	Y	Y	82
Blankenship et al. [63]												P ^1^

Y = Yes; N = No; U = Unclear; NA = Not applicable; % = Percentage of score; *: assessed primary and secondary outcome. ^1^ There is no a critical appraisal tool for a general literature review. Therefore, both reviewers decided to include the study and rated P (*pass*) for synthesis.

**Table 3 ijerph-17-06889-t003:** Risk of Bias Assessment within the studies (*n* = 29).

Study	Random Sequence Generation (Selection Bias)	Allocation Concealment (Selection Bias)	Blinding of Participants and Personnel (Performance Bias)	Blinding of Outcome Assessment (Detection Bias)	Incomplete Outcome Data (Attrition Bias)	Selective Reporting (Reporting Bias)	Other
Aasa et al. [39]	Low risk	Unclear risk	Low risk	Low risk	Low risk	Low risk	Low risk
Abdallah et al. [40]	Low risk	Unclear risk	Low risk	Low risk	Low risk	Low risk	Low risk
Abdallah et al. [41]	Low risk	Unclear risk	Low risk	Low risk	Low risk	Low risk	Low risk
Azmi et al. [51]	Low risk	Low risk	Unclear risk	Low risk	Low risk	Low risk	Low risk
Bahramnezhad et al. [52]	Low risk	Low risk	Unclear risk	Low risk	Low risk	Low risk	Low risk
Bakhai et al. [53]	Low risk	Low risk	Unclear risk	Low risk	Low risk	Low risk	Low risk
Benzer et al. [42]	Low risk	Low risk	Unclear risk	Low risk	Low risk	Low risk	Low risk
Blankenship et al. [63]	Low risk	Unclear risk	Unclear risk	Low risk	Low risk	Low risk	Low risk
Borkon et al. [29]	Low risk	Low risk	Low risk	Low risk	Low risk	Low risk	Low risk
Bourassa et al. [55]	Low risk	Unclear risk	Unclear risk	Low risk	Low risk	Low risk	Low risk
Chudek et al. [56]	Low risk	Low risk	Unclear risk	Low risk	Low risk	Low risk	Low risk
Cohen et al. [43]	Low risk	Low risk	Low risk	Low risk	Low risk	Low risk	Low risk
Favarato et al. [26]	Low risk	Unclear risk	Unclear risk	Low risk	Low risk	Low risk	Low risk
Kim et al. [44]	Low risk	Low risk	Low risk	Low risk	Low risk	Low risk	Low risk
Kim et al. [45]	Low risk	Low risk	Low risk	Low risk	Low risk	Low risk	Low risk
Koch et al. [57]	Low risk	Unclear risk	Unclear risk	Low risk	Low risk	Low risk	Low risk
Koltowski et al. [28]	Low risk	High risk	High risk	Low risk	Low risk	Low risk	Low risk
Krzych et al. [62]	Low risk	Low risk	Unclear risk	Low risk	Low risk	Low risk	Low risk
Li et al. [46]	Low risk	Low risk	Unclear risk	Low risk	Low risk	Low risk	Low risk
Rinfret et al. [47]	Low risk	Low risk	Low risk	Low risk	Low risk	Low risk	Low risk
Schenkeveld et al. [15]	Low risk	Low risk	Unclear risk	Low risk	Low risk	Low risk	Low risk
Sipotz et al. [58]	Low risk	Unclear risk	Unclear risk	Low risk	Low risk	Low risk	Low risk
Sjoland et al. [59]	Low risk	Unclear risk	Unclear risk	Low risk	Low risk	Low risk	Low risk
Spertus et al. [60]	Low risk	Unclear risk	Unclear risk	Low risk	Low risk	Low risk	Low risk
Takousi et al. [49]	Low risk	Unclear risk	Unclear risk	Low risk	Low risk	Low risk	Low risk
Tegn et al. [48]	Low risk	Low risk	Low risk	Low risk	Low risk	Low risk	Low risk
Veenstra et al. [50]	Low risk	Unclear risk	Unclear risk	Low risk	Low risk	Low risk	Low risk
Wahrborg et al. [33]	Low risk	Low risk	Low risk	Low risk	Low risk	Low risk	Low risk
Yan et al. [61]	Low risk	Unclear risk	Unclear risk	Low risk	Low risk	Low risk	Low risk

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
