# Peer review of "Quality of Life Changes in Acute Coronary Syndromes Patients: A Systematic Review and Meta-Analysis"

_ijerph, 2020, doi:10.3390/ijerph17186889_

Round 1

Reviewer 1 Report

This manuscript describes an attempt to compile data addressing the impact of treatment upon quality of life after treatment for unstable coronary artery disease. The authors readily admit that the task was daunting, the methods used in various studies highly varied and the outcome of this analysis questionable. There appear to be some useful observations here but the manuscript deserves additional work. Particularly, the question addressed, or perhaps that should be addressed, is how a QoL score may be weighed against evidence that compels specific treatment options. It seems that the underwhelming change in scores in some instances may relate to treatment driven by perceived threat to life rather than direct impact on quality of life and that, unlike the treatment of cancer, the treatment related impairment is often minor and fleeting.

  1. Line 46. A concept is introduced with no definition. What is HRQoL. In addition, the reference does not claim that HRQoL is a predictor of outcome, rather that rehabilitation alters quality of life. 
  2. Line 63. What is does, "the impact QoL in decision making of therapy prescription" mean? I think that I understand the concept that you are trying to describe, but your description is unclear.
  3. Line 71 is nonsensical. You have not stated a study design at all. Your design becomes evident at line 89,90. Perhaps this line (71) should just be deleted.
  4. Line 127. Why was geography or study origin not included among confounders. Cultural and geographic differences in populations substantially impact upon the acceptance of CABG surgery or PCI. Patient expectations from treatment impact upon perceived change in quality of life. Therefore, if some cultures eschew surgery, PCI will be applied more liberally and sometimes when surgery may provide a better probability of long term success.
  5. Page 8. Table. Takousi: Inclusion of the information from this reference suggests that you included a meta-analysis in your meta-analysis. Is this not a potential introduction of error? The same question arises from Blankenship on page 9, a review article.
  6. Line 214. As part of your results, you are quoting a review article rather than a data finding.
  7. Line 232. Isn't this the most important finding. The change in quality of life score is partly determined by the success or failure of the treatment, but equally or greater by the baseline degree of impairment and expectations of the patient. In several of the quoted studies, patients entered treatment with a "good" quality of life score or symptoms score, making improvement quite difficult.
  8. The argument (288-301) that QoL should be included in a patient's assessment and treatment choice is laudable. However, the choices taken for the treatment of ACS are driven primarily by acuity, anatomy and the goals of survival, attenuation of myocardial infarction severity and lastly by symptom relief.
  9. Line 339. Is the fact that people felt better after being treated for ACS really a discovery? In the following statements, you bring to bear a statement that you discounted as unreliable when it was presented in your results: specifically that quality of life change differed between patients treated with PCI and surgery. This does not belong in your conclusion.
  10. Do the authors think that patients who were quoted their potential risk of death or disability before and after the performed procedure might provide different results in a quality of life study?

Author Response

Quality of Life Changes in Acute Coronary Syndromes Patients: a Systematic Review and Meta-Analysis

Reviewer 1:

This manuscript describes an attempt to compile data addressing the impact of treatment upon quality of life after treatment for unstable coronary artery disease. The authors readily admit that the task was daunting, the methods used in various studies highly varied and the outcome of this analysis questionable. There appear to be some useful observations here but the manuscript deserves additional work. Particularly, the question addressed, or perhaps that should be addressed, is how a QoL score may be weighed against evidence that compels specific treatment options. It seems that the underwhelming change in scores in some instances may relate to treatment driven by perceived threat to life rather than direct impact on quality of life and that, unlike the treatment of cancer, the treatment related impairment is often minor and fleeting.

Thank you for feedback and we have addressed all the comments.

  1. Line 46. A concept is introduced with no definition. What is HRQoL. In addition, the reference does not claim that HRQoL is a predictor of outcome, rather that rehabilitation alters quality of life. 

Thanks for the observation. We have added the definition of HRQoL: “a pluri-dimensional concept which used to assess the impact of health status on quality of life via multiple indicators comprising physical and emotional functioning, well-being and alleviation of symptoms.” In relation to the second point, we quote the relevant text on page 2 of Anchah et al. 2017: “HRQoL is an independent predictor of mortality and morbidity in patients who are suffering from post-acute coronary syndrome (ACS)”. We have also paraphrased this line as follows:  “Anchah and colleagues posit that, in patients who are suffering from post-acute coronary syndrome (ACS), HRQoL is an independently predictor of mortality and morbidity) [10]..

  1. Line 63. What is does, "the impact QoL in decision making of therapy prescription" mean? I think that I understand the concept that you are trying to describe, but your description is unclear.

Thanks, we make it clearer mow. “………the impact of QoL on physicians’ decision making in relation to therapeutic prescription …….”

  1. Line 71 is nonsensical. You have not stated a study design at all. Your design becomes evident at line 89,90. Perhaps this line (71) should just be deleted.

Thank you, this section has now been deleted

  1. Line 127. Why was geography or study origin not included among confounders. Cultural and geographic differences in populations substantially impact upon the acceptance of CABG surgery or PCI. Patient expectations from treatment impact upon perceived change in quality of life. Therefore, if some cultures eschew surgery, PCI will be applied more liberally and sometimes when surgery may provide a better probability of long term success.

As indicated in Table 1, we have included the study origin for every study. The list on line 127 was not exhaustive as we listed just some of the confounders that were included in as secondary outcomes.

  1. Page 8. Table. Takousi: Inclusion of the information from this reference suggests that you included a meta-analysis in your meta-analysis. Is this not a potential introduction of error? The same question arises from Blankenship on page 9, a review article.

One of the rationales of this systematic review is to provide the most up to date information. Following best practice guidelines from JBI and others, it was important that this relevant systematic review be included in the study. In terms of introduction of error in our metanalysis, the reviewer will note that while Takousi et al was included in Table 1, this study was not included in the metanalysis (please see Figure 2).  

  1. Line 214. As part of your results, you are quoting a review article rather than a data finding.

As pointed in our response to comment 5 from this reviewer, including and summarising information from relevant systematic reviews within subsequent is acceptable and appropriate practice. Please see section. Please refer to answer #5.

  1. Line 232. Isn't this the most important finding. The change in quality of life score is partly determined by the success or failure of the treatment, but equally or greater by the baseline degree of impairment and expectations of the patient. In several of the quoted studies, patients entered treatment with a "good" quality of life score or symptoms score, making improvement quite difficult.

Thank you for this observation. We have edited this test as follows: “A significant finding in this review was differed according to patients’ clinical and non-clinical characteristics: sex, age and ACS diagnosis”

We have also included the difficulty associated with ascribing QoL as pointed out by this reviewer as a limitation on page 26: “It is possible that the small changes in HRQoL scores seen in some of the studies included in our review could have been due to treatment driven by perceived threat to life rather than direct impact on quality of life. The inclusion, within the metanalysis, of many RCTs may have however reduced this confounding”.

  1. The argument (288-301) that QoL should be included in a patient's assessment and treatment choice is laudable. However, the choices taken for the treatment of ACS are driven primarily by acuity, anatomy and the goals of survival, attenuation of myocardial infarction severity and lastly by symptom relief.

Thank you.  We have noted this observation and amended the manuscript accordingly.

  1. Line 339. Is the fact that people felt better after being treated for ACS really a discovery? In the following statements, you bring to bear a statement that you discounted as unreliable when it was presented in your results: specifically that quality of life change differed between patients treated with PCI and surgery. This does not belong in your conclusion.

The findings in the paper showed improved QoL after treatment, which confirms the previous findings. However, these improvements are different by treatment and duration, which we have described in the following statement—these are all findings from the included papers.

  1. Do the authors think that patients who were quoted their potential risk of death or disability before and after the performed procedure might provide different results in a quality of life study?

We did not find any evidence on this in the review. We therefore did not include any information about this inquiry from the reviewer.

Reviewer 2 Report

This review is interesting; it covers an important topic that is becoming increasingly important in quality of life in acute coronary syndromes patients. The work is very well designed and well written. The only minor thing that I wanted to recommend is to put space (n=xx to n = xx, see also the flowchart and please unify the style of citations).

Author Response

Quality of Life Changes in Acute Coronary Syndromes Patients: a Systematic Review and Meta-Analysis

Reviewer 2:

This review is interesting; it covers an important topic that is becoming increasingly important in quality of life in acute coronary syndromes patients. The work is very well designed and well written. The only minor thing that I wanted to recommend is to put space (n=xx to n = xx, see also the flowchart and please unify the style of citations).

Thank you. We have included this.

Reviewer 3 Report

This is an interesting review that suggests the quality of life acute coronary syndrome patients improves after receiving treatment and that this improvement appears to vary by type of treatment. Overall, the analysis was rigorous and adhered to standard methodology, including careful categorization of bias. My comments are mostly minor.

1) QoL is a multi-dimensional concept that covers many classic domains such as physical, mental, emotional, and social functioning, with fatigue, pain, emotional distress, social activities and roles being important factors. Assessing the positive aspects or well-being of one’s life (e.g., positive emotions and life satisfaction) also is often a component of QoL assessment, as well as participations measures to account for specific factors such as vision loss, mobility difficulty, and intellectual deficits that may impact QoL. Given the multifaceted aspect of assessing QoL it is important to discuss such differences (strengths and limitations) given that the various studies included in this review were not homogeneous in their assessment.    

2) The nature of the procedure also is important to consider, for example emergent/urgent vs. elective CABG surgery. Perhaps some details can be provided in the tables. 

3) Most diseases are the result of the interaction between many innate and many environmental factors and ACS is no exception. However, cherry-picking is a common complaint of meta-analysis. Please suggest otherwise.

4) Studies finding significant effects are more likely to be published than studies finding marginal or no effects (i.e., file drawer problem). Please counter argue this in the case of current manuscript.

5) Line 24. Better to change the word “confirmed” to “suggested”.

6) Line 26. Please spell out first usage of HRQoL.

Author Response

Quality of Life Changes in Acute Coronary Syndromes Patients: a Systematic Review and Meta-Analysis

Reviewer 3:

This is an interesting review that suggests the quality of life acute coronary syndrome patients improves after receiving treatment and that this improvement appears to vary by type of treatment. Overall, the analysis was rigorous and adhered to standard methodology, including careful categorization of bias. My comments are mostly minor.

Thank you, we have addressed your minor comments

1) QoL is a multi-dimensional concept that covers many classic domains such as physical, mental, emotional, and social functioning, with fatigue, pain, emotional distress, social activities and roles being important factors. Assessing the positive aspects or well-being of one’s life (e.g., positive emotions and life satisfaction) also is often a component of QoL assessment, as well as participations measures to account for specific factors such as vision loss, mobility difficulty, and intellectual deficits that may impact QoL. Given the multifaceted aspect of assessing QoL it is important to discuss such differences (strengths and limitations) given that the various studies included in this review were not homogeneous in their assessment.    

Thank you very much. We have added these points in the definition of HRQoL. We also appreciate this in our limitation section.

2) The nature of the procedure also is important to consider, for example emergent/urgent vs. elective CABG surgery. Perhaps some details can be provided in the tables. 

Thank you very much. We have provided some details on this to the context of ‘duration and the complexity of the surgery’ and the quality of life.

3) Most diseases are the result of the interaction between many innate and many environmental factors and ACS is no exception. However, cherry-picking is a common complaint of meta-analysis. Please suggest otherwise.

Thanks and we appreciate this. Given this is existing fact we prefer not to add.

4) Studies finding significant effects are more likely to be published than studies finding marginal or no effects (i.e., file drawer problem). Please counter argue this in the case of current manuscript.

 it’s evident that negative findings are discouraged by publishers and even affect authors from reporting these findings confidently. Thanks you but this is not the case and even not the aim of this study.

5) Line 24. Better to change the word “confirmed” to “suggested”.

 Done! Thank you.

6) Line 26. Please spell out first usage of HRQoL.

Thank you but we have already described in full in line 11.